# Platelet-Rich Plasma for Pleurodesis: An Experimental Study in Rabbits

**DOI:** 10.3390/medicina58121842

**Published:** 2022-12-15

**Authors:** Styliani Maria Kolokotroni, Dimitrios Lamprinos, Nikolaos Goutas, Emmanouil I. Kapetanakis, Konstantinos Kontzoglou, Despoina Perrea, Periklis Tomos

**Affiliations:** 1Department of Cardiothoracic Surgery, University Hospitals Coventry and Warwickshire NHS Trust, Coventry CV2 2DX, UK; 2Laboratory of Experimental Surgery and Surgical Research “N. S. Christeas”, Medical School, National and Kapodistrian University of Athens, 11527 Athens, Greece; 3Emergency Department, Laiko General Hospital, 11527 Athens, Greece; 4Department of Forensic Medicine and Toxicology, Medical School, National and Kapodistrian University of Athens, 11527 Athens, Greece; 5Department of Thoracic Surgery, Attikon University Hospital, Medical School, National and Kapodistrian University of Athens, 12462 Athens, Greece; 6Second Department of Propedeutic Surgery, Laiko General Hospital, Medical School, National and Kapodistrian University of Athens, 11527 Athens, Greece

**Keywords:** PRP, pleurodesis, rabbits, talc

## Abstract

*Background and Objectives*: This study was designed to evaluate platelet-rich plasma (PRP) as a method of pleurodesis in a rabbit model. Pleurodesis with PRP was compared against the gold-standard use of talc. The secondary evaluation assessed the ideal time for achieving pleurodesis. *Materials and Methods*: 25 healthy New Zealand white rabbits were assigned to three groups, as follows: 12 animals in the first and second groups, as well as one animal with no intervention in the final group, which was used as a control. The talc pleurodesis group (baseline) underwent pleurodesis with sterile talc, which is the gold-standard sclerosing agent used for pleurodesis. The PRP group underwent pleurodesis using autologous PRP. The last group had one rabbit with no intervention. A total of 12 rabbits (*n* = 6 for the talc pleurodesis group and *n* = 6 for the PRP group) were sacrificed 3 days (72 h) after the intervention, and 12 rabbits (*n* = 6 for the talc pleurodesis group and *n* = 6 for the PRP group) were sacrificed 6 days (144 h) after the intervention. In both the talc and PRP group, FBC and CRP were measured before the intervention and in 3 or 6 days afterwards, respectively. The pleura and the lungs were evaluated histopathologically. *Results*: Macroscopically, there were no statistically significant differences between the two groups. In terms of microscopic findings, there were no statistically significant differences in inflammatory reactions provoked in the visceral and parietal pleura between the PRP and talc. In addition, with talc pleurodesis, a foreign-body reaction was observed in about 50% of the cases, which was not observed with PRP. In terms of inflammation between 3 and 6 days, there were no statistically significant differences with PRP, there was only a statistically significant difference between 3 and 6 days regarding the parietal pleura in the talc group. *Conclusions*: The instillation of autologous PRP in the pleural cavity shows promise in achieving pleurodesis. The efficacy of PRP as a pleurodesis agent should be examined further.

## 1. Introduction

The most common cause of recurrent pleural effusion is malignancy. Malignant pleural effusions (MPEs) affect more than 1 million people every year. The annual incidence in the US alone is over 150,000 patients per year, and in the UK, the annual incidence is about 40,000 [1]. MPEs can affect up to 15% of all patients with cancer, and their incidence is likely to increase as the cancer incidence increases overall and survival improves. The median survival time after diagnosis is 3–12 months depending on the individual patient and the specific tumour factors [2].

Up to 75% of patients with MPE develop symptoms such as dyspnoea, cough, and chest pain. The dyspnoea associated with pleural effusion can lower the quality of life of a cancer patient with a life expectancy of <6 months [3].

Chemical pleurodesis is one of the most common methods used to prevent recurrence and can significantly improve a patient’s quality of life by reducing mainly dyspnoea and, less commonly, cough and chest pain, even though it has not been demonstrated to improve survival [4].

Pleurodesis can also be used for the treatment of recurrent secondary pneumothorax [5,6,7].

Pleurodesis is considered if there is adequate lung expansion, otherwise if the lung is trapped, the treatment of choice is the placement of an indwelling pleural catheter.

It consists of the instillation of a sclerosing agent in the pleural space with the aim to achieve symphysis of the parietal and visceral pleura and thus obliterate the potential pleural space through inflammation and/or adhesion formation.

It has been suggested that the successful pleurodesis may offer a survival benefit in patients with MPE, irrespective of the type of primary cancer [8].

There is a variety of sclerosing agents that have been tried in order to achieve pleurodesis, such as talc, silver nitrate, antibiotics, antineoplastic drugs, antimalarials, and immunomodulators. The success rates vary from 54% to 93% and these depend on the agent used and possibly on the dose and means of treatment.

Talc is the gold-standard agent, as it is widely available, inexpensive, and highly efficient. However, both minor and major side effects are associated with talc pleurodesis. The minor side effects include fever, chest pain, and wound infections. The major side effects include empyema, hypotension, dysrhythmia, acute respiratory distress syndrome (ARDS 0.7–9%), acute lung injury (ALI), and death 0.9–2.5% [9,10].

Platelet-rich plasma (PRP) is a portion of the plasma of autologous blood that contains platelets in a concentration above the baseline (before centrifugation). It contains not only a high number of platelets but also all the clotting factors and a range of growth factors, chemokines, and cytokines, and has mitogenic, angiogenic, and chemotactic properties.

The biologically active substances mentioned above are responsible for initiating the coagulation cascade and creating new connective tissue and revascularisation. PRP can induce proliferation, migration, cellular differentiation, and angiogenesis. It has the advantage of being an autologous product and is safe without known side effects [11].

It is obtained from the blood of the patients before centrifugation. After centrifugation, the blood is separated in three layers: red blood cells, PRP, and platelet-poor plasma. PRP has been used in a variety of medical specialties such as haematology, maxillofacial surgery [12], cardiac surgery, paediatric surgery, gynaecology, urology, plastic surgery, ophthalmology, dentistry, dermatology [13], orthopaedic surgery, spinal surgery, and neurosurgery [14,15]. It has been used in a variety of cases in thoracic surgery as well.

In our study, we compared PRP against the gold-standard talc to achieve pleurodesis in rabbits. We evaluated pleurodesis microscopically by assessing the inflammation and fibrosis provoked with each agent used.

## 2. Materials and Methods


**Animals and study design:**


We used rabbits for our experiment, as they are phylogenetically more similar to humans than other animals, such as rodents [16].

A total of 25 healthy male New Zealand white rabbits that were 3 months old and weighed 3020 g ± 305 g (mean and SD values, respectively) were used with the approval of the Ethical Committee of the University of Athens Medical School. The surgical procedures, as well as the preoperative and postoperative care, were performed in accordance with the ethical standards.

The rabbits were assigned to three groups of 12 animals in the first and second group, as well as one animal with no intervention in the third group, which was used as a control.

The talc pleurodesis group (baseline) underwent pleurodesis with sterile talc in a dose of 100 mg/kg. The PRP group underwent pleurodesis using autologous PRP. The last group had one rabbit with no intervention, which was used as a control animal for the injury caused by the surgical procedures. A total of 12 rabbits (*n* = 6 for the talc pleurodesis group and *n* = 6 for the PRP group) were sacrificed in 3 days (72 h) after the intervention and 12 rabbits (*n* = 6 for the talc pleurodesis group and *n* = 6 for the PRP group) were sacrificed in 6 days (144 h) after the intervention.

We chose to sacrifice the animals in 3 days initially to correspond to the same time frame that is believed to achieve pleurodesis in humans when the chest drain is usually removed. We then doubled the time to sacrifice rabbits at 6 days as a pilot study. As the preliminary results were encouraging, we decided to include more animals to be sacrificed at 6 days. As this is a small study, we felt it was best to analyse these results first and then to proceed with a larger scale study in the future.


**Surgical procedure:**


The substance that would be instilled in the pleural cavity was prepared first.

In the PRP group, 17.2 mL of whole blood was drawn from the marginal ear vein. The blood was collected with 4 syringes containing citrate with a capacity of 4.3 mL each. These syringes were then centrifuged in a Hettig centrifuge at 4000 rpm for 6 min at room temperature. By using these particular devices, 1.2–3 mL of autologous PRP was derived from each animal. In the talc group, talc was administered at a dose of 100 mg/kg corresponding to approximately 3 mL of solution, using a Steritalc (Novatech SA, Orama Medical, Athens, Greece) 4 g vial diluted in 50 mL saline (0.9% NaCl).

The rabbits were anaesthetised lightly with 5 mg/kg ketamine hydrochloride (Imalgene, Merial, Lyon, France) plus 25 mg/kg xylazine hydrochloride (Rompun, Bayer Animal Health GmbH, Leverkusen, Germany) administered intramuscularly. In both the talc and PRP group, before any intervention, 5 mL of blood was drawn from the marginal ear vein and FBC and CRP were measured. The thorax was prepared for aseptic surgery by shaving the right chest. Then, the rabbits were intubated using a V-Gel large laryngeal mask for rabbits (Docsinnovent Ltd, Hemel Hempstead, UK). The V-gel mask was connected to the ventilator in the volume control mode and set to 25 breaths per minute. For some animals, more administration of ketamine and xylazine was required to achieve the desired level of anaesthesia.

The rabbits were positioned in a left lateral decubitus position and the right chest wall was scrubbed with povidone-iodine 7.5% (Betadine, Lavifarm A.E., Paiania, Greece). A 1 cm incision was performed in the mid axillary line in the 5th intercostal space using a scalpel with an 11 blade. The subcutaneous tissue and the muscle were dissected, and by breaching the pleura, the right pleural space was entered. Under direct vision, a 20-gauge catheter connected to a 5 mL syringe was inserted in the pleural cavity. Depending on the group the rabbit belonged to, either PRP or sterile talc were instilled in the pleural cavity under sterile conditions. After the instillation of either PRP or talc, a Nelaton 14 Ch catheter was inserted in the pleural cavity. A Vicryl 2-0 suture was placed around the catheter. This was followed by the application of positive pressures and a Valsalva test using an Ambu bag to prevent pneumothorax. After ensuring that the underlying lung was expanded and confirming that there was no air leakage, the Nelaton catheter was removed and the incision was closed with Vicryl 2-0 sutures, both for the underlying tissues (muscle and subcutaneous tissue) and the skin.

The left hemithorax received no intervention and served as a control.

At the end of the procedure, the rabbits were given a shot of enrofloxacin (Baytril, Bayer Animal Health GmbH, Leverkusen, Germany). At the moment the rabbits could breathe spontaneously, and the level of consciousness was satisfactory, the laryngeal mask was removed.

After the surgery, the rabbits were monitored closely for the next hour in recovery for signs of pain or respiratory distress and then they were returned to their enclosures. They received food and water in the afternoon.

Every day thereafter they received a daily injection of enrofloxacin (Baytril, Bayer Animal Health GmbH, Leverkusen, Germany) intramuscularly.

The rabbits were monitored daily for sepsis, respiratory distress, pain, or any other complication. Betadine was applied to the wound daily to prevent wound infection.

The rabbits were sacrificed after 3 or 6 days, respectively, depending on the group they belonged to.

Before the rabbits were sacrificed, 5 mL of blood was drawn from the marginal ear vein and FBC and CRP were measured. The rabbits were anaesthetised lightly with 5 mg/kg ketamine hydrochloride (0.8 mL) (Imalgene, Merial, Lyon, France) and 25 mg/kg xylazine hydrochloride (0.8 mL) (Rompun, Bayer Animal Health GmbH, Leverkusen, Germany) administered intramuscularly. After this, 5 ml of theopentobarbital (Dolethal, Vetoquinol UK Limited, Towcester, UK) was administered intravenously to the marginal ear vein.

The right (with talc or PRP) and the left hemithoraces (no substance instilled) were dissected and assessed for any evidence of macroscopic pleurodesis. They were removed separately, as follows: thoracic wall en block with the parietal pleura; lungs with the visceral pleura. Subsequently, the specimens were submerged in a 10% formalin solution and were sent for histopathological analysis to the Toxicology Department of Athens Medical School.

The specimens were processed routinely. Samples of the parietal pleura en block with the chest wall and the visceral pleura en block with the lung, were obtained from 4 to 5 randomly selected sites from each hemithorax. The samples were stained with haematoxylin and eosin (H&E) and Masson’s trichrome for collagen. They were examined by classic optic microscopy using an Olympus BX43 and lenses Olympus Plan CN 2.5× 10× 20× 40× 60×. All slides were evaluated by the same senior pathologist, who is an associate professor of histopathology, to mitigate any potential observer bias.

The microscopic slides were evaluated for the presence of inflammation and fibrosis (cellularity, neovascularity and mesothelial cell proliferation). The degree of inflammation was graded from 0 to III for absent, mild, moderate, and severe, respectively, judging by the quantity of inflammatory cells present. With 0 corresponded to <5% inflammatory cells, I was 6–33%, II was 34–65% and III was 66–100%. The fibrosis was assessed as well using the same grading of 0-III, as reflected in the Masson-Trichrome stain. The greater the positivity of the Masson stain quantitatively, the more the grading increases. This grading is in accordance with other similar studies in the literature [5].

The blood test results were evaluated as a secondary parameter. 


**Statistical analysis:**


All the laboratory data points were expressed as mean ± SD. The results after the instillation of talc or PRP were compared after 3 or 6 days, in order to assess whether there were any statistical differences in WBC and subtypes, platelets, CRP, and Hb using the two-tailed t-test if the data met the criteria for normality. The degree of inflammation was divided in two subsets depending on the microscopic findings; the first subset consisted of the degree of inflammation from 0 to I and the second subset of II to III. Fisher’s exact test was used to determine whether there was a statistically significant difference between all groups (PRP and Talc; 3 days and 6 days). The differences in results were considered significant when *p* < 0.05. The statistical analysis was performed using the SPSS v26 software package (IBM, New York, NY, USA).

## 3. Results

A total of 25 rabbits were included in the experiment. A total of 12 rabbits were sacrificed at 3 days and 12 rabbits were sacrificed at 6 days. In each group, six rabbits had talc instillation into the right thoracic cavity and six had PRP, respectively; the rabbit with no intervention was the last group examined. During the course of our experiment, six rabbits died in the process; these were not included in the results and were replaced. Interestingly, the autopsy performed in one of the rabbits that passed away demonstrated severe bronchopneumonia on the contralateral lung, which was not related to our interventions. Rabbits are prone to chest infections irrespective of any intervention; this specific rabbit died on the table just after our intervention and it was felt that the bronchopneumonia was likely pre-existing.

The amount of talc used in experiments with rabbits varies in the literature, starting at 50 mg/kg to the maximum dosage of 400 mg/kg. We chose to administer 100 mg/kg. which is the closest to the human equivalent of 70 mg/kg, making it comparable to human pleurodesis in clinical practice.

The amount of PRP extracted and used reflected the human equivalent extraction of PRP for other purposes. As this is an original study, we had no previous studies to reference for guidance.

The rise of CRP and WBC (inflammatory markers) post-talc pleurodesis has been demonstrated in the past and may suggest a systemic inflammatory response; however, this was not demonstrated in our measurements [9,17].

We originally hypothesised that PRP could achieve a pleurodesis comparable to talc; therefore, both agents were investigated. We also wanted to assess whether there were any further changes noticed in the pleura by doubling the time of examining the pleura.

There was no visible pleural fluid or any evidence of macroscopic pleurodesis (fibrin strands/adhesions or thickened pleura) in any specimen examined, either in the talc group, the PRP group, or the 3-day or 6-day group. In the talc group, some rabbits had visible collections of talc ranging between 1 and 5 mm, mostly in the dependent areas of the lung (ventral), but as this was sporadic and random, it was not included in our analysis.

The results of the microscopic examinations of the right pleura are shown in Figure 1, Figure 2, Figure 3 and Figure 4 with talc and PRP, respectively, both on H&E and Masson’s Trichrome stain.

Where talc was administered, light microscopy showed sporadic focal inflammatory responses and the presence of giant cells—a foreign body reaction in the lung of the rabbits in at least half of the specimens. The main histologic changes observed were the mesothelium denudement, disorganisation of basal lamina and underlying connective tissue, vessel congestion, haemorrhagic infiltrates, and focal fibrin deposition, both in the parietal and visceral pleura.

In addition, the underlying pulmonary parenchyma showed alveolar wall thickening, capillary vasodilation and hyperaemia, phenomena associated with interstitial oedema, and intense leukocyte infiltration.

Histologic studies revealed that lymphocytes and macrophages were the main infiltrated cell types. There was only a small number of neutrophils. The redistribution of these infiltrated cells was observed late after instillation.

Where PRP was administered, mesothelium denudement, disorganisation of basal lamina and underlying connective tissue, vessel congestion, haemorrhagic infiltrates, and focal fibrin deposition were observed in both in the parietal and visceral pleura. The underlying lung showed alveolar wall thickening, capillary vasodilation, hyperaemia, and interstitial oedema. There was no foreign body reaction observed in any of the specimens.

Fibrosis was evaluated with a Masson trichrome stain for collagen. This was not significant in any of the groups examined, ranging in the area of 0 to I, absent to mild.

The left lung, with no intervention, was examined as well and did not show any inflammation or fibrosis.

The statistical analysis demonstrated:

Regarding the blood samples taken, as shown on Table 1:When comparing blood samples before and after intervention between PRP and talc:
○Regarding WBC, RBC, Hb, Ht, MCV, MCH, MCHC, platelets, RTW CV, MPV, neutrophils, lymphocytes, baseophils, and CRP, there was no statistically significant difference between the two groups.○There was a statistically significant difference between the two groups for monocytes (*p* = 0.049) and eosinophils (*p* = 0.011).

Regarding achieving pleurodesis, judging by the inflammation provoked in the visceral and parietal pleura:When comparing the two groups of PRP and talc, the results using Fisher’s exact test demonstrated that there was no statistically significant difference between inflammation with PRP and talc, both regarding visceral and parietal pleura (*p* = 0.217 and *p* = 0.4). (Figure 5 and Figure 6).

When comparing pleurodesis between 3 and 6 days for both groups (PRP and talc combined), the results using Fisher’s exact test demonstrated:
○Regarding visceral pleura, there was no statistically significant difference between the two groups (*p* = 0.217) (Figure 7).○Regarding parietal pleura, there was a statistically significant difference between the two groups (*p* = 0.009) (Figure 8).

Subdividing 3 and 6 days for PRP and talc (using Fisher’s exact test)
○with PRP: there are no statistically significant differences between 3 and 6 days both for the visceral and the parietal pleura (*p* = 0.182 and *p* = 1, respectively) (Figure 9 and Figure 10).○With talc, there was a statistically significant difference in the inflammation provoked in 6 days compared to 3 days regarding parietal pleura (*p* = 0.002) (Figure 11); regarding visceral pleura, this could not be measured, as inflammation was high in both groups (grade II–III).

The univariate analysis for inflammation is shown on Table 2.

## 4. Discussion

The most common indication for pleurodesis is pleural effusion. Usually, the effusion is malignant, secondary to end-stage cancer. The patients are usually elderly. Life expectancy in these patients is limited and simple drainage cannot prevent the re-accumulation of the effusion. Treatment with a chest drain alone is therefore usually not sufficient and pleurodesis is advocated.

The ideal agent for chemical pleurodesis should be of low cost, available worldwide, easy to administer, safe to use, and well tolerated with minimal or no side effects. It should also have a high molecular weight and chemical polarity, low regional clearance, rapid systemic clearance, and a steep dose-response curve.

The gold-standard agent for pleurodesis is considered to be talc, as it is effective to achieve pleurodesis, widely available, inexpensive, and has been demonstrated to be more effective than other sclerosing agents, with a success rate around or more than 90%, irrespective of the method and the dose given [18,19,20,21]. Talc (Mg_3_Si_4_O_10_[OH]_2_) was first used as a sclerosing agent in 1935 [22]. Talc used intrapleurally is sterilised and asbestos-free. It has a success rate of achieving pleurodesis of up to 93% either via slurry or poudrage. It is used for the management of both pleural effusion and pneumothorax [23,24]. It is usually well tolerated, with chest pain (pleuritic) and mild fever being the most common side effects. The other acute events reported include hypotension, dyspnoea, fainting, hypoxaemia and, less commonly, hypercalcaemia [25]. However, there are concerns about its safety, with ARDS appearing in post-talc pleurodesis in up to 9% of patients, leading to respiratory failure and an overall mortality of 1% in some series [26]. In patients who developed ARDS post-talc instillation in the thoracic cavity, talc particles were detected in the bronchoalveolar lavage fluid and lung tissues. In the study by Dressler et al. [27] comparing talc insufflation at a thoracoscopy with the talc slurry, the mortality related to the treatment was reported in 9 of 242 patients in the talc insufflation group and in 7 of 240 patients in the talc slurry group. The most important risk factors for the development of ARDS appear to be the dosage, size, and type of talc used [28,29]. Rossi et al. suggested that talc insufflation using larger particles is safer than talc insufflation with small particles [30]. A recent study by Shinno et al. suggested that old age and the presence of interstitial abnormalities on the chest CT may present risk factors for the development of ARDS after pleurodesis with large particle size talc [31].

Several studies suggest that ARDS in the talc pleurodesis occurs due to pulmonary talc deposition. Talc can be found in the lungs of any patient who undergoes talc pleurodesis, not only in those who develop ARDS. The dissemination of talc extrapleurally has been demonstrated in animals and humans [32,33]. Werebe et al. suggested that talc particles are rapidly absorbed through the pleura, which can be demonstrated in bronchoalveolar lavage and in the lung [32]. Karsner et al. suggested that talc moves from the pleura to the parietal pleural lymphatics and then to the mediastinal lymph nodes and the thoracic duct, from where it enters the systemic circulation [34]. Therefore, it is preferred that patients with benign disease are not treated with talc. Montes et al. suggested that the higher the dose of talc administered, the greater the chance of developing fibrotic visceral pleural thickening and foreign-body granulomas, which could lead to unwanted acute and chronic inflammatory responses [29]. However, currently most talc preparations are standardised [25].

A recent meta-analysis of the complications of thoracoscopic talc insufflation revealed that pain and fever were the most frequent complications. ARDS was one of the less observed complications. Pneumothorax and pneumonia were also among the most highly prevalent. Emphysema, prolonged drainage, prolonged air leak and thromboembolism had moderate incidence rates. Respiratory failure, lung injury, re-expansion pulmonary oedema, pulmonary embolism, empyema, and arrhythmia were among the least frequently observed complications [35]. Anaphylactic shock has been mentioned in case reports [36].

The search for the ideal agent for pleurodesis, therefore, continues.

Sterile talc, silver nitrate, polidocanol (a long-acting local anaesthetic that has been used to treat varicose veins as well for its thrombotic and sclerosant effect) [37], doxycycline, erythromycin [38], and a combination of talc and doxycycline [39] have been used in rabbits and appears to be effective in achieving pleurodesis in various experiments. More recently, the pleurodesis with biomaterial implants has been tried as well and appears to be effective in experiments in rabbits, as hydrogel and non-hydrogel foam formulations along with talc [40,41] and poly-ε-caprolactone (PCL) gel for pneumothorax [42].

Platelet-rich plasma (PRP) is a biological product that is a portion of the plasma of autologous blood that contains platelets in a concentration above the baseline (before centrifugation). It is also known as platelet-rich fibrin (PRF) matrix, platelet-rich growth factors and platelet concentrate.

It contains not only a high number of platelets but also all the clotting factors; a range of growth factors, with the most important ones being the platelet-derived growth factor (PDGF); transforming growth factor β (TGF-β); insulin growth factor-1 (IGF-1); vascular endothelial growth factor (VEGF); epidermal growth factor (EGF); hepatocyte growth factor (HGF); fibroblast growth factor (FGF); chemokines and cytokines; and has mitogenic, angiogenic, and chemotactic properties.

It is obtained from the blood of the patients before centrifugation. After centrifugation, the blood is separated into three layers: red blood cells, PRP and platelet-poor plasma. Another thing to consider is whether PRP contains leucocytes or not.

There are several commercial devices available to produce PRP. A concentration of platelets 2–5 times above the baseline can be achieved.

PRP is used with the rationale that an injection of concentrated platelets can promote tissue repair by releasing biologically active substances such as growth factors, cytokines, lysosomes, and adhesion proteins, which are responsible for initiating the coagulation cascade, creating new connective tissue and revascularisation. PRP can induce the release of growth factors above normal levels to start the healing process in trauma. It can induce proliferation, migration, cellular differentiation, and angiogenesis. PRP has the advantage of being an autologous product and being safe without known side effects [11].

The terminology PRP was first used in the 1970s by haematologists. It was initially used as a transfusion product for patients with thrombocytopaenia. In the following decade, PRP started being used in maxillofacial surgery and then for sports injuries and became increasingly popular when used by famous athletes [12]. Other medical specialties that use PRP are cardiac surgery, paediatric surgery, gynaecology, urology, plastic surgery, ophthalmology, dentistry, dermatology [13], orthopaedic surgery, spinal surgery, and neurosurgery [14,15].

In thoracic surgery, PRP has been used in a variety of cases. PRP can enhance healing and reduce complications in tracheal anastomosis by inducing angiogenesis and releasing growth factors in an experiment with pigs [43]. A similar experiment with rabbits used PRP to enhance the laryngotracheal anastomosis [44]. PRP has been successful in treating tracheobronchial fistulas post anatomical lung resection in humans [45] and in treating massive haemoptysis by injecting it to the diseased area on the bronchus [46], as well as in treating chylothorax post transhiatal oesophagectomy with fibrin sheets for pleurodesis [47] and in enhancing the staple line post bullectomy for pneumothorax along with PGA sheets [48]. It has been used in rabbits to promote rib cartilage regeneration [49] and the healing of rib fractures [50]. It has been used to enhance the bronchial stump post-pneumonectomy in pigs [51] and to help healing combined with negative pressure wound therapy on sternal osteomyelitis and sinus tract after thoracotomy [52].

PRP has been also used in rabbits for bone regeneration [53], added to polypropylene meshes to facilitate angiogenesis and integration, for urinary incontinence [54], to enhance sternal healing along with gelatin hydrogel [55], and to regenerate cartilage in large osteochondral defects [56].

The mechanisms that induce pleurodesis are not completely understood.

It is likely that pleurodesis is a generic process and nonspecific to the sclerosant. The necessary condition for pleurodesis to be successful is for the visceral and parietal pleura to have a complete apposition.

The biological mechanisms for pleurodesis involve the same ultimate pathways, including diffuse inflammation, activation of pleural cells, and the coagulation cascade, which plays an instrumental role in the whole process, with decreased fibrinolytic activity and increased fibrinogenesis, fibrin chain formation, fibroblast recruitment and proliferation, and the production of collagen and extracellular matrix components. [57,58].

Pleurodesis undoubtedly relates to an injury to the pleura, which triggers the inflammatory cascade (through cytokines such as IL-8), angiogenesis (i.e., production of the vascular endothelial growth factor) and fibrogenesis (transforming growth factor-β), as well as reduces the activities of the fibrinolytic system, all of which eventually result in the development of pleural adhesions and fibrosis [20,22].

It has been hypothesised that the initial event is an acute injury to the pleura [5], which must be significant. Diffuse inflammation with an acute inflammatory reaction occurs early, in day 1. The extent, intensity and duration of the inflammation will guide the end result, as the administration of corticosteroids inhibit talc pleurodesis in rabbits [59], and the intrapleural instillation of TNF-α blocking antibodies can also reduce the degree of talc pleurodesis [60]. The outcome of pleurodesis in humans may be affected by the administration of steroids systemically, which could also be dose-dependent [61].

It has also been hypothesised that the intrapleural injection of a sclerosing agent causes injury to the mesothelial cells, ranging from a cuboidal transition to total desquamation. In the first 24 h, there is a denudement of the mesothelium. Initially, damage to the mesothelium and deposition of fibrin in the pleural space and connective tissue occur. There is increasing evidence that the pleural mesothelium is the primary initiator of the mechanisms leading to pleurodesis. For satisfactory results, the sclerosing agent must reach the maximum surface area of the normal mesothelium in the pleural space. This might explain that the rate of failure is higher if the mesothelium is covered by the tumour.

This probably results in the secretion of chemokines, such as the interleukin-6, and 1β, (IL-6, IL-1β), interleukin 8 (IL-8) and monocyte chemoattractant protein (MCP-1), as well as growth factors—vascular endothelial growth factor (VEGF), platelet-derived growth factor (PDGF), basic fibroblast growth factor (bFGF), and transforming growth factor- β (ΤGF-β) [62,63,64]. The intact mesothelial cells with the above cytokines are instrumental in the initiation and maintenance of different pathways of pleural inflammation and pleural space obliteration [65]. The mesothelial cell injury produces fibronectin, which is important in fibrin formation and eventual pleural fibrosis. The mesothelial cells secrete collagen, metalloproteinases that degrade collagen and inhibitors of the metalloproteinases [66]. It is likely that pleurodesis is developed when the procollagen factors are predominant instead of the anticollagen factors, which likely induce normal tissue repair. It is also possible that inflammatory cells from the blood stream, such as neutrophils and mononuclear phagocytes, play an important role in pleurodesis.

Regeneration of the damaged cells and migration of connective tissue cells to the damaged area occur afterwards (days 3 to 5), followed by the production of extracellular matrix proteins (days to several weeks) and collagenisation, with the regaining of wound strength, a process that starts in the first week and carries on for several weeks [67]. There is a progressive thickening of the pleura in 7 days with adherent fibrin in areas of mesothelial denudement and base membrane injury, and a slow resolution of the mononuclear predominant pleural inflammation, as the fibroblast proliferate and collagen is deposited [68].

Collagen is likely to be the most important protein in achieving pleurodesis. It is produced primarily in fibroblasts. The first step is the production of collagen propeptides or α-chains. The propeptides are intracellular and undergo several modifications and aggregates to form procollagen molecules, which are still soluble and have three-chain helices. Thereafter, they are excreted from the cells and form tropocollagen, which is insoluble. Cross-linkages between α-chains of adjacent tropocollagen molecules occur and thin immature collagen fibres are formed. These mature by reacting with other fibres and lose water to become thick mature collagen fibres. These are responsible for the resistance of the visceral pleura to stretching, which is believed to be instrumental to pleurodesis.

This then leads to pleural fibrosis and fusion of the visceral and parietal pleura.

To summarise pleurodesis mechanisms, the mesothelial cells of the pleura are the primary target for the sclerosant and play a fundamental role in the whole pleurodesis process, including diffuse inflammation, pleural coagulation-fibrinolysis imbalance (favouring the formation of fibrin adhesions), fibroblast recruitment, and subsequent proliferation and production of collagen after intrapleural application of the sclerosing agent. When the tumour burden is high, normal mesothelial cells are scarcely present, and then the response to the sclerosing agent is lower, leading to the failure of pleurodesis. Complications associated with pleurodesis might be minimised using large-particle talc [57].

Experiments with rabbits 7 days after the talc slurry have demonstrated pleural thickening, fibrin deposition where the mesothelium has been denuded, and a transient mononuclear vasculitis [69,70].

Our study is the first study to our knowledge that uses PRP as a pleurodesis agent. In our study, we compared PRP as a pleurodesis agent against the gold standard pleurodesis agent, which is talc. Our study demonstrated that pleurodesis is achievable with PRP, as there was no difference in the inflammatory reaction provoked in the visceral and parietal pleura when PRP or talc were used. Pleurodesis, measured as the inflammation provoked, was more prominent in 6 days compared to 3 days in the talc subset but not in the PRP subset.

We believe that the reason no gross pleurodesis was observed was that the time of animal sacrifice was less than in other studies, so adhesions did not have time to form. In various studies, adhesions are observed after 28 days of treatment, at least [40].

There are limitations that apply to our study. First, the animals were sacrificed in 3 and 6 days, while animals in other studies to evaluate pleurodesis are sacrificed in 14 or 28 days. As our study is a small study, we felt it was best to analyse the preliminary results first and then to perform a larger scale study in the future, also adding image analysis algorithms. A suggested next step would be to sacrifice the animals in one month after the instillation of PRP compared to talc, to assess for macroscopic adhesions and fibrosis.

Second, although rabbits bear many similarities to humans, they do have a thinner pleura compared to humans; therefore, one must be careful when interpreting the results. However, these results can be used as a guidance. Third, this is an experimental study with a small number of animals.

## 5. Conclusions

In conclusion, the results of this study demonstrate that PRP is effective in achieving pleurodesis in an experimental model in rabbits. There was no statistically significant difference in achieving pleurodesis between PRP and talc, which is the gold-standard pleurodesis agent. Additionally, there were no side effects or foreign body reactions observed with PRP. The autologous PRP demonstrates no side-effects.

The efficacy of PRP as a pleurodesis agent should be examined further in animal studies where the animals are sacrificed more than 6 days after the experiment, ideally a month, at least.

PRP can be further evaluated as a pleurodesis agent in humans using randomised controlled studies. Meanwhile, the search for the ideal agent for pleurodesis continues.

## Figures and Tables

**Figure 1 medicina-58-01842-f001:**
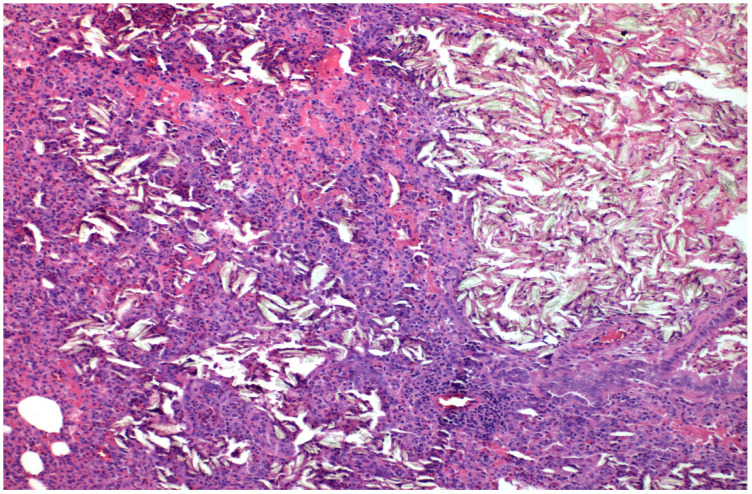
×100 Hematoxylin-Eosin stain. Foreign Body Crystal Aggregations after talc installation and Pulmonary Parenchymal Thickening.

**Figure 2 medicina-58-01842-f002:**
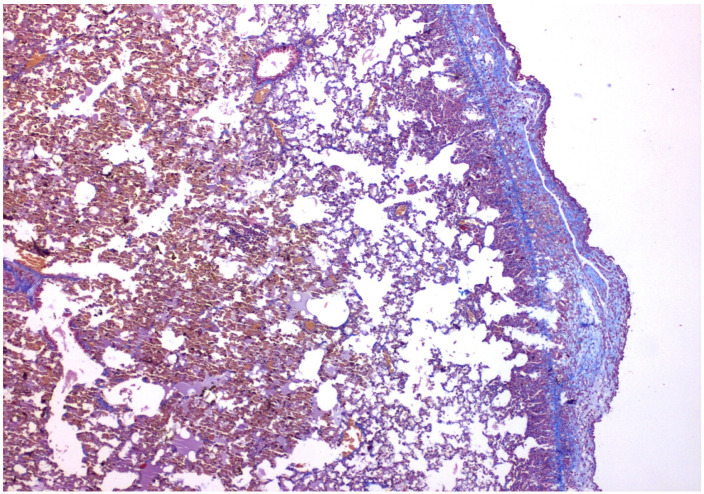
×40 Masson-Trichrome stain: Following talc installation, fibrosis is present under the mesothelial layer.

**Figure 3 medicina-58-01842-f003:**
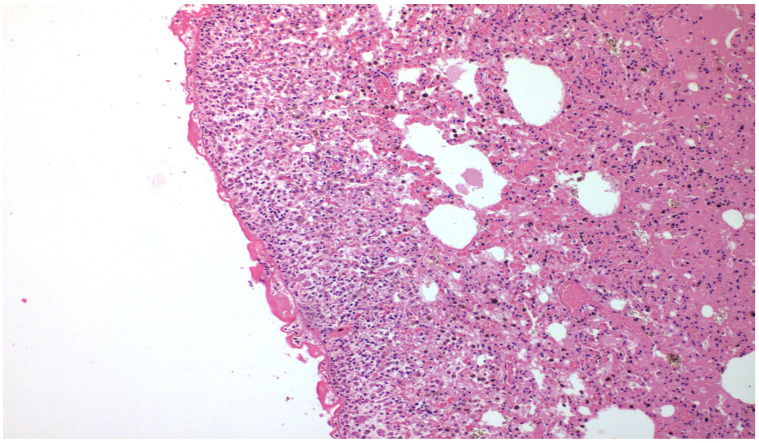
×100 Hematoxylin-Eosin stain—after PRP installation, there is significant fibrosis peripherally.

**Figure 4 medicina-58-01842-f004:**
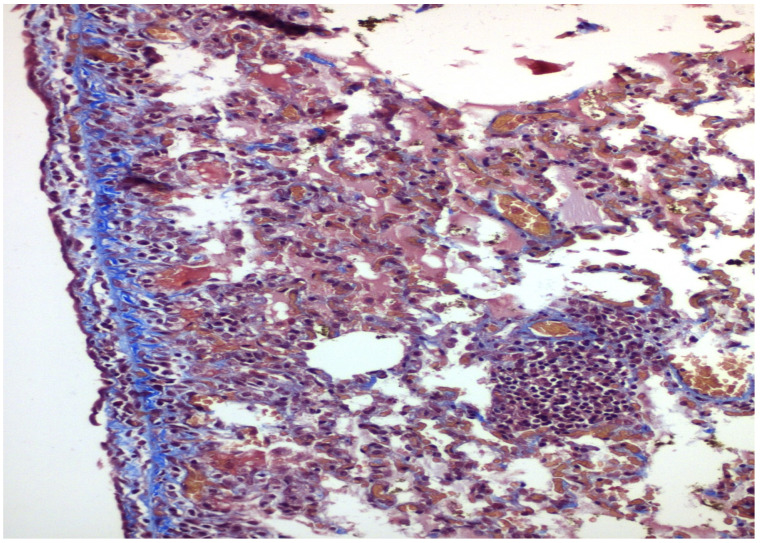
×200 Masson-Trichrome stain: Following PRP installation, fibrosis is present under the mesothelial layer.

**Figure 5 medicina-58-01842-f005:**
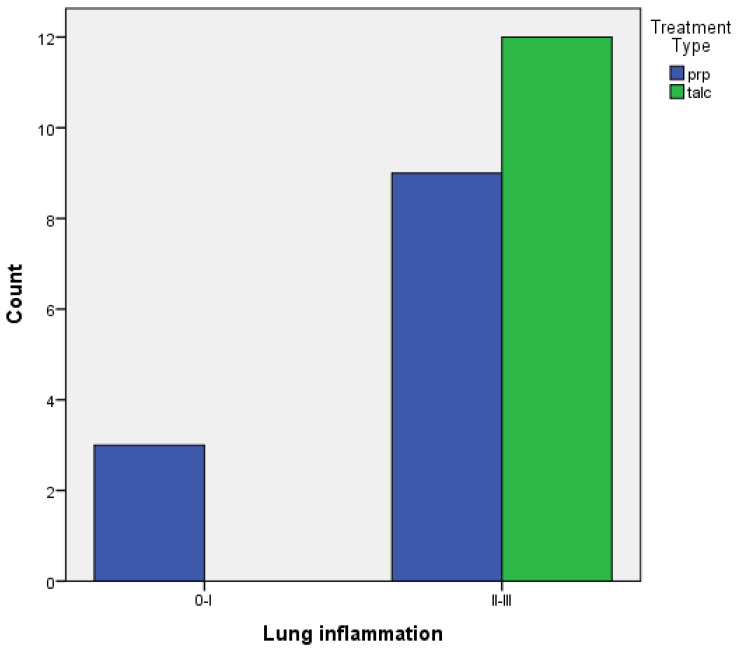
Visceral pleura inflammation of PRP vs. talc.

**Figure 6 medicina-58-01842-f006:**
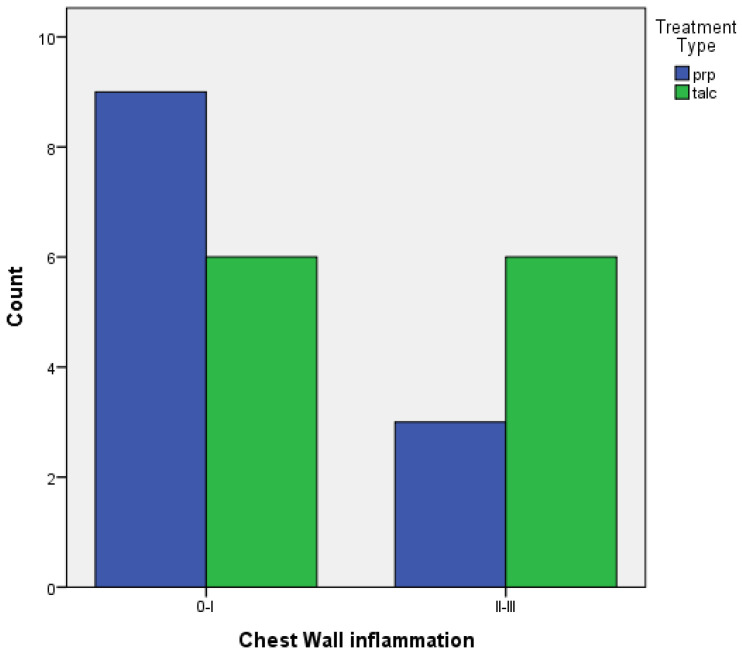
Parietal pleura inflammation of PRP vs. talc.

**Figure 7 medicina-58-01842-f007:**
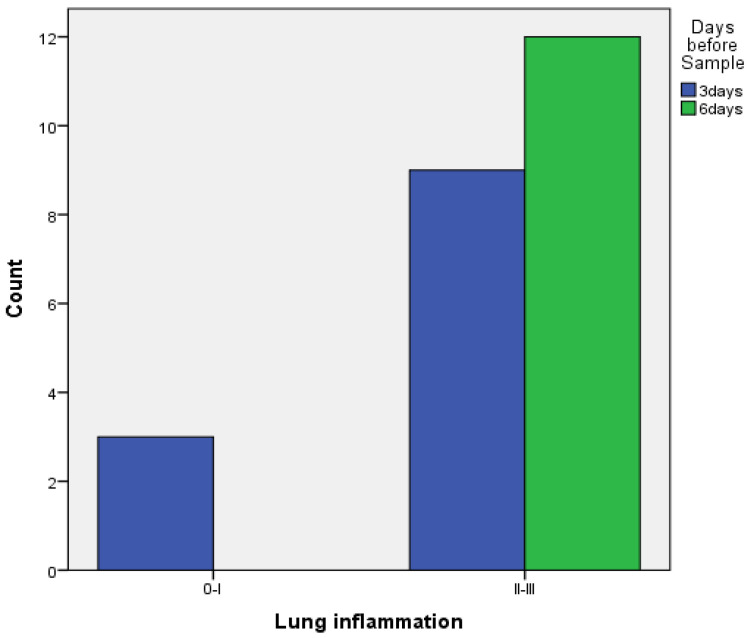
Visceral pleura inflammation at 3 vs. 6 days of PRP and talc combined.

**Figure 8 medicina-58-01842-f008:**
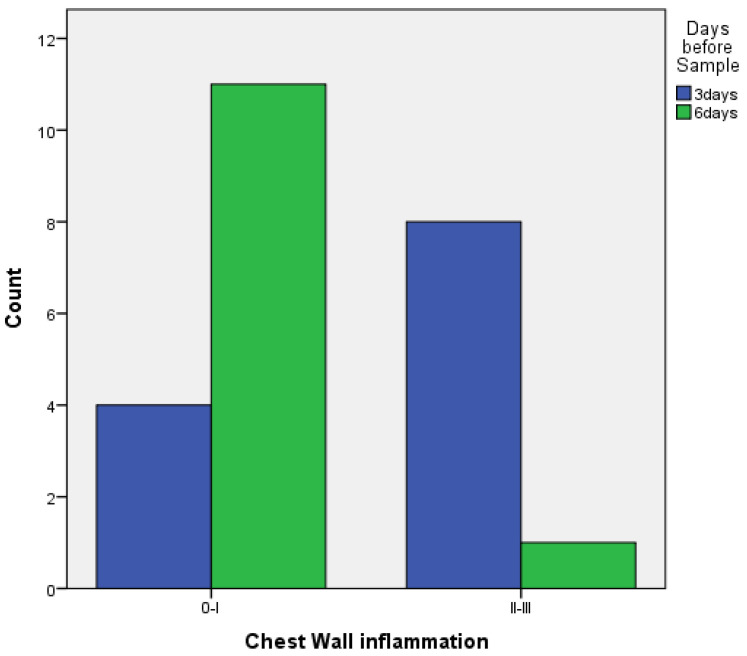
Parietal pleura inflammation at 3 vs. 6 days of PRP and talc combined.

**Figure 9 medicina-58-01842-f009:**
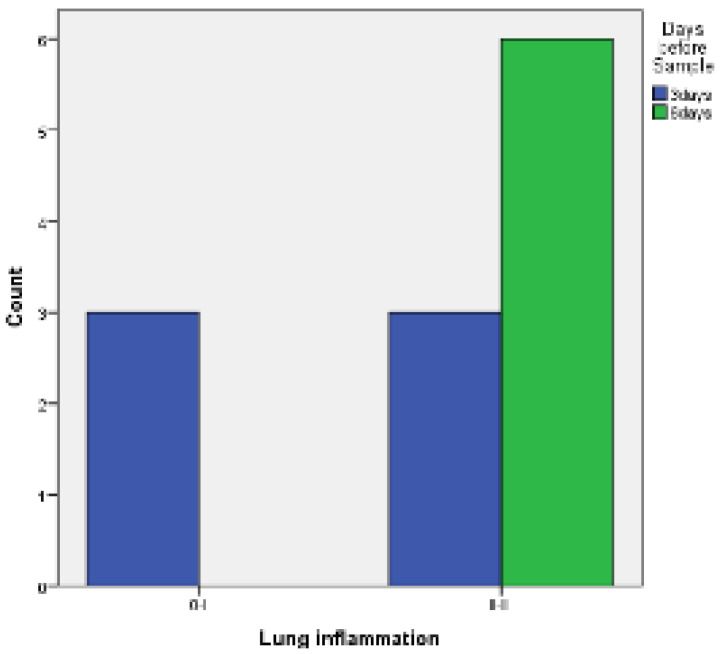
Visceral pleura inflammation 3 vs. 6 days of PRP.

**Figure 10 medicina-58-01842-f010:**
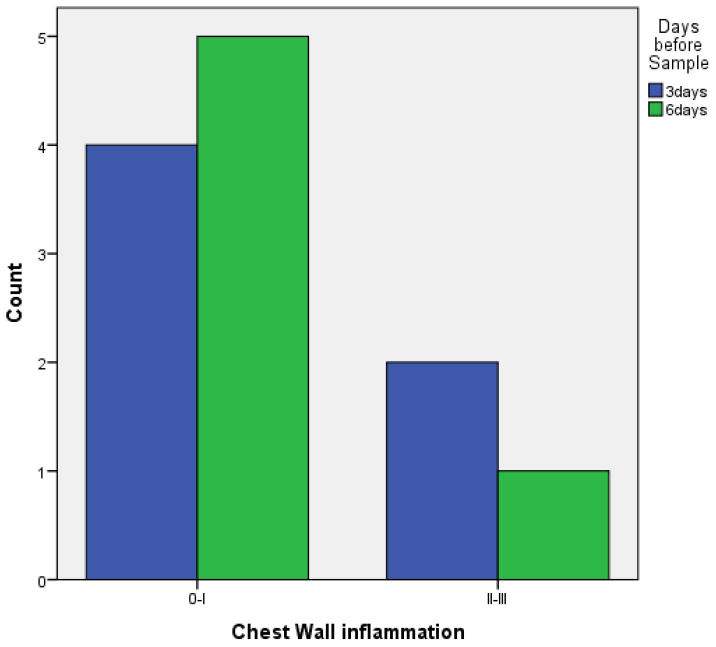
Parietal pleura inflammation 3 vs. 6 days of PRP.

**Figure 11 medicina-58-01842-f011:**
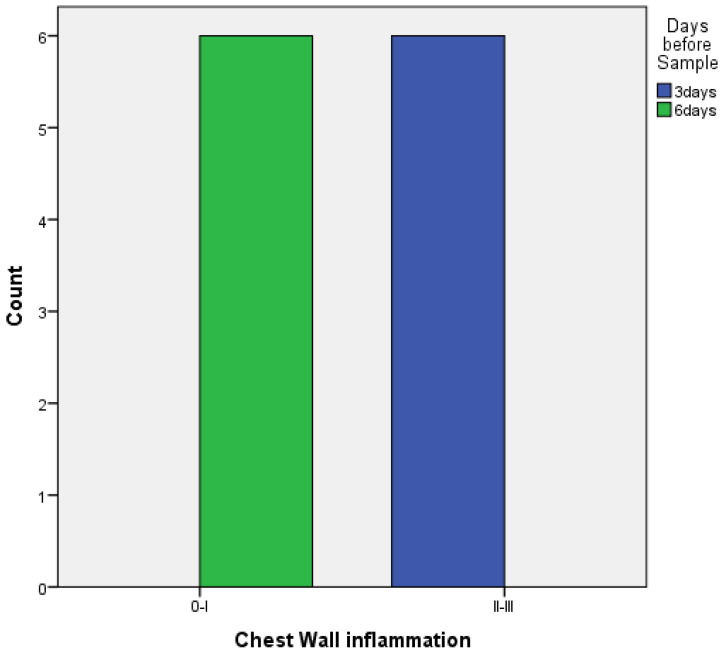
Parietal pleura inflammation at 3 vs. 6 days of talc.

**Table 1 medicina-58-01842-t001:** Blood test results comparing PRP and talc post-intervention values (mean ± SD).

Blood Test	PRP	Talc	*p*-Value
WBC	6957 (±2246)	6587 (±1824)	0.701
RBC	5,623,000 (±295,787)	5,602,222 (±564,942)	0.874
Hb	10.91 (±0.45)	11.02 (±0.43)	0.58
Platelets	276,700 (±101,793)	368,888 (±184,243)	0.19
Neutrophils	40.12 (±14.3)	42.44 (±14.26)	0.73
Lymphocytes	52.81 (±15.75)	41.9 (±14.75)	0.14
Monocytes	6.18 (±3.46)	11.86 (±7.69)	0.049
Eosinophils	0.85 (±0.79)	3.33 (±2.64)	0.011
Baseophils	0.05 (±0.16)	0.42 (±1.27)	0.37
CRP	<0.03	<0.03	

**Table 2 medicina-58-01842-t002:** Univariate analysis for inflammation.

Inflammation	PRP(%)	TALC(%)	*p*-Value
**Pleura**			0.4
0–I	9 (60%)	6 (40%)	
II–III	3 (33.3%)	6 (66,7%)	
**Lung**			0.217
0–I	3 (100%)	0	
II–III	9 (42.9%)	12 (57.1%)	

## Data Availability

Not applicable.

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
