# Peer review of "Platelet-Rich Plasma for Pleurodesis: An Experimental Study in Rabbits"

_medicina, 2022, doi:10.3390/medicina58121842_

Round 1

Reviewer 1 Report

This is a well written paper on an experimental study of pleurodesis using animal models. The authors experiment with a novel method of pleurodesis using platelet-rich plasma in a small group of healthy rabbits. The rabbits were sacrificed 3 and 6 days after the procedure.

The study design is mostly correct for the specified purpose. The authors describe their methodology in detail, enabling replication of their findings. Results are presented clearly and logically. The discussion is comprehensive and focuses on the pathophysiology of pleurodesis, framing their results within current theoretical knowledge.

However, the last point, which is an advantage of this publication is also necessary to make it meaningful. Histological and serological markers of inflammation are used as surrogates for pleurodesis, given the short time between the procedure and sacrifice. It is interesting that this time frame was chosen; was it because of logistical reasons? What was the rationale?

The authors demonstrate that PRP can induce acute inflammation of the pleura, mesothelial denudement and fibrin deposition, overlapping with talc, minus the foreign body reaction, as expected. Arguing that these serve as markers of pleurodesis is not wrong, but it is optimistic. Subtle differences in the inflammatory process cannot be discerned on light microscopy alone and there is no guarantee that pleurodesis would actually have resulted from these procedures. As such, what this study demonstrates is that indeed, PRP can induce acute inflammation in the pleura that is similar in the evaluated characteristics to talc induced inflammation. I believe this objective should be stated in the introduction, perhaps using other words, leading the way into the methods and creating a better flow to the paper.

The study relies heavily on histological findings and blood test results. I would argue that the former are more specific than the latter for the specified objective. Given their importance, the methodology underlying histopathological findings needs to be better fleshed out. Why did you use polarizing microscopy with H&E and M. Trichrome stains? The provided photographs were not taken with polarized light. What does a grade of inflammation grade? What does a grade of fibrosis mean? These are subjective measurements. Were the histology slides all evaluated by the same observer? What is this observer's histopathology background? Ideally, objective measurements, using image analysis algorithms, would be obtained. However, this can be cumbersome for such a small project. My suggestion is that the authors mitigate these problems by refining the methods and developing a visual analogue scale of the grades. This should prove useful to guide further research.

There is one inconsistency between results and methods. The authors state that "Histologic studies revealed that lymphocytes, mostly T lymphocytes and macrophages 238 were the main infiltrated cell types"; however, T lymphocytes cannot be distinguished visually from other types of lymphocytes on the stains that were used. Did the authors perform immunohistochemistry for CD3? If so, this should be stated in the methods.

Some other smaller notes:

-On lines 94 and 95, the expense of the animal models are not really relevant as as justification for choosing rabbits

-On lines 191 to 193, do you have an hypothesis as to why bronchopneumonia developed in the contralateral lung?

-On line 211, consider replacing findings with analysis.

-Graphics and tables have minor incorrections, and could be made more visually appealing to potential readers

This publication should lead to follow-up studies using this methodology, providing the basis for future research. Looking forward to your feedback.

Author Response

Thank you for your review and your comments.

In terms of your comments:

In terms of the time frame chosen, 3 and 6 days: we chose to sacrifice the animals in 3 days initially as it is believed that in humans, pleurodesis (with talc) is achieved in 3 days when the chest drain is usually removed.  We then doubled the time to sacrifice rabbits at 6 days as a pilot study. As the preliminary results were encouraging, we decided to include more animals to be sacrificed at 6 days. As this is a small study, we felt it was best to analyse these results first and as they were encouraging,  to carry on with a larger scale study in the future.

In similar papers, examining pleurodesis with talc in rabbits, pleurodesis was evaluated microscopically by assessing the degree of inflammation and fibrosis. We followed a similar pattern. We could add in the introduction as a last paragraph: In our study we compared PRP against the gold-standard talc to achieve pleurodesis in rabbits. We evaluated pleurodesis microscopically by assessing the inflammation (and fibrosis) provoked with each agent used.

The blood test results were only processed as a secondary parameter and not as a marker of pleurodesis.

We used a classic optic microscope and this will be amended accordingly in the text.

 Both inflammation and fibrosis were graded as 0-III in accordance with other studies in the literature.

Grade of inflammation: this is related to the quantity of inflammatory cells present

Grade of fibrosis: this is reflected at Masson Trichrome stain. The greater the positivity of Masson stain quantitatively, the more the grading increases.

The histology slides were evaluated by the same histopathologist who is a senior associate professor of histopathology. This is  a small project. Given the encouraging initial results, we are looking forward to expanding our experiment further with image analysis algorithms as you suggest.

I will delete mostly T lymphocytes on 238 and amend the phrase to: Histologic studies revealed that lymphocytes and macrophages were the main infiltrated cell types. Immunohistochemistry with C3 antibodies was tried in our pilot study but did not yield evaluable results due to tissue adsorption of the chromogen.

-On lines 94 and 95, the expense of the animal models are not really relevant as as justification for choosing rabbits: Agreed, I will remove this.

-On lines 191 to 193, do you have an hypothesis as to why bronchopneumonia developed in the contralateral lung? Rabbits are prone to chest infections irrespective of any intervention, this specific rabbit died on the table just after our intervention and it was felt that the bronchopneumonia was likely pre-existing and not related to our intervention. I have included this change in the paper

-On line 211, consider replacing findings with analysis: thank you, it will be changed in the revised version

-Graphics and tables have minor incorrections, and could be made more visually appealing to potential readers: I have amended the alignment and sizing of the graphics/tables.  

This publication should lead to follow-up studies using this methodology, providing the basis for future research. Looking forward to your feedback.

In conclusion thank you for your comprehensive comments and positive feedback. I sincerely hope this publication does lead to follow up studies using this methodology.

Reviewer 2 Report

I congratulate the authors. I have read this paper with great interest and found it well-conducted and written. As the authors already suggested, the main limit of the study is the short time between pleurodesis and the autopsy. It is an exciting pilot study overall. However, I would suggest extending the observation to 1 month after the instillation of PRP/talc, to look for any macroscopic adhesions and to achieve a potential impact on clinical research. Despite this limit, I recommend this paper for publication. 

Please check for minor spelling errors. e.g. in line 63, the word instillation is misspelled. 

Author Response

I congratulate the authors. I have read this paper with great interest and found it well-conducted and written. As the authors already suggested, the main limit of the study is the short time between pleurodesis and the autopsy. It is an exciting pilot study overall. However, I would suggest extending the observation to 1 month after the instillation of PRP/talc, to look for any macroscopic adhesions and to achieve a potential impact on clinical research. Despite this limit, I recommend this paper for publication. 

Many thanks for your review, hopefully this publication will lead to further follow-up studies over a longer duration. I have included your remarks in the limitations of the study.

Please check for minor spelling errors. e.g. in line 63, the word instillation is misspelled. Noted, I have amended.